# Vulvovaginal Candidosis: Current Concepts, Challenges and Perspectives

**DOI:** 10.3390/jof6040267

**Published:** 2020-11-07

**Authors:** Valentina Sustr, Philipp Foessleitner, Herbert Kiss, Alex Farr

**Affiliations:** Department of Obstetrics and Gynecology, Division of Obstetrics and Feto-Maternal Medicine, Medical University of Vienna, Waehringer Guertel 18–20, A-1090 Vienna, Austria; n1306881@students.meduniwien.ac.at (V.S.); philipp.foessleitner@meduniwien.ac.at (P.F.); herbert.kiss@meduniwien.ac.at (H.K.)

**Keywords:** *Candida albicans*, diagnosis, epidemiology, fungal infection, review, vulvovaginal candidosis

## Abstract

Vulvovaginal candidosis (VVC) is a frequently occurring infection of the lower female genital tract, mostly affecting immuno-competent women at childbearing age. *Candida albicans* is the most prevalent pathogenic yeast—apart from other non-*albicans* species—related to this fungal infection. Different virulence factors of *C. albicans* have been identified, which increase the risk of developing VVC. To initiate treatment and positively influence the disease course, fast and reliable diagnosis is crucial. In this narrative review, we cover the existing state of understanding of the epidemiology, pathogenesis and diagnosis of VVC. However, treatment recommendations should follow current guidelines.

## 1. Background

Vulvovaginal candidosis (VVC) is a disease that affects women of all ethnicities and social classes [1,2]. The exact epidemiology of this disease is based on unreliable data, although based on what is currently known, 70–75% of women experience VVC at least once in their lifetime [2,3]. Numerous risk factors for VVC have been described. However, the pathological mechanism underlying the progression from colonization to infection is not fully understood [2], partly because most cases do not present a specific trigger [4,5]. The estrogenized vagina of asymptomatic women is often colonized with *Candida* species, although colonization does not necessarily lead to an infection [6]. The predominant species in 90–95% of cases is *Candida albicans*, followed by non-*albicans* species, such as *C. glabrata*, *C. tropicalis*, *C. krusei* and *C. parapsilosis* [7]. If VVC is caused by non-*albicans* species, it mostly manifests as a mild infection [4].

VVC causes distress in many affected women, and it is the reason for many consultations in gynecological offices worldwide. Herein, we summarize different aspects of this disease, excluding treatment approaches and concepts, using a non-systematic methodology to search the existing literature.

## 2. Epidemiology

Epidemiological data on the incidence of VVC are relatively vague because studies are often imprecise or conducted in unrepresentative populations; the incidence is estimated to range from 12% to 57% in the overall female population [2,3]. An Estonian study that used barcoded pyrosequencing technology found *Candida* in 67.6% of the asymptomatic women, reporting that the mycobiome that colonizes the healthy vaginal environment is more diverse than what was previously recognized [8]. Table 1 summarizes the VVC incidence and *Candida* colonization among symptomatic and asymptomatic women in different countries [3,9,10,11,12,13].

Apart from asymptomatic fungal colonization, *Candida*-related infections are the second leading cause of vaginitis, primarily affecting women during their reproductive lifetime, when high estrogen levels increase the glycogen content of the vaginal epithelium, therewith playing a role in the nutrition for the yeast [3,14]. It is of paramount importance to differentiate between colonization and infection because 50% of women with an infection will experience a second episode, and 5–8% will develop recurrent vulvovaginal candidosis (RVVC) [2,15]. RVVC is defined as experiencing four or more episodes of VVC per year [2,16]. Recent data report a worldwide RVVC prevalence of approximately 138 million women per year and an additional 372 million over a lifetime [4,5]. Most episodes of RVVC occur between the ages of 19–35 years, and according to a survey, the prevalence rate is 9% by the age of 50 years [5,17]. Fidel et al. [18,19] suggest that women with RVVC have a dysfunction in the normal protective immune response acquired from a previous *Candida* infection.

## 3. Pathogenesis

Recognition of the importance of the innate response in driving inflammatory responses associated with VVC has led to many recent insights regarding the pathogenesis of VVC. It is known that cells of the innate host express receptors that recognize pathogen-associated molecular patterns (PAMPs) [20]. The major classes of receptors that recognize *Candida*-associated molecular patterns are those of the Toll-like receptors and lectin-like receptors families [21]. A mannose-binding lectin (MBL) recognizes and binds to *Candida* surface mannan, increases complement activation and inhibits *Candida* growth [22,23]. Similarly, macrophages, dendritic cells and epithelial cells play an important role and exert a protective function. Macrophages and dendritic cells have specific surface receptors that recognize MBL and promote opsonization of MBL-bound microorganisms [24].

Adaptive immunity—including fungus-specific defense mechanisms—is developed directly or indirectly through cell-mediated immunity (T-cells) [25]. Decreased T-cell-mediated immunity is associated with increased susceptibility to VVC in women who are immunocompromised (e.g., due to HIV infection, previous organ transplantation, glucocorticoid therapy or antineoplastic chemotherapy) [26,27,28]. B-cells and immunoglobulin-secreting plasma cells migrate into the vaginal epithelium [18,29], protecting against pathogenic microorganisms. The exact mechanism by which antibodies protect against *Candida* is unknown [30,31]. Immunoglobulin A (IgA) and IgG are the predominant immunoglobulin classes found in vaginal secretions [32].

During the pathogenic process, *Candida* undergoes reversible yeast-to-hyphae transition, which causes changes in the type of surface carbohydrates, affecting the adhesion and invasion of vaginal epithelial cells (Figure 1). *Candida* either infects vaginal epithelial cells directly through the invasion of hyphae or indirectly through the contact of PAMPs with pattern recognition receptors (PRRs). In response to this, contact inflammatory immune mediators—chemokines, cytokines, antimicrobial peptides or damage-associated molecular patterns—are secreted, subsequently recruiting innate immune cells, such as macrophages, dendritic cells and neutrophils. These cells also recognize PAMPs through PRRs on their surfaces, bind to the pathogen, and stimulate its removal by phagocytosis.

This mechanism involves the production of reactive oxygen species, which further regulate all stages of inflammation. The phagocytized pathogenic components activate inflammasomes, which induce the release of proinflammatory cytokines, and subsequently promote T-cell activation and neutrophil recruitment. Neutrophils enter through the vaginal epithelium and promote phagocytosis of *Candida* on the epithelial surfaces of the vagina [33,34,35].

For *Candida* colonization on the host surface, adhesion to vaginal epithelial cells is crucial and contributes to infection and persistence [36]. This process is promoted by cell-surface components (adhesins), which recognize host ligands, such as serum proteins, in the extracellular matrix of host tissues (e.g., laminin, fibronectin, collagen, vitronectin and entactin) [37]. A major group of adhesins from the agglutinin-like sequence (ALS) gene family is encoded in *C. albicans*; this group consists of 8 members (ALS 1–7, ALS 9) [38]. Cheng et al. [39] found that ALS 1–3 and ALS 9 occurred more frequently in women with VVC.

### 3.1. The Role of Enzymes

*Candida* spp. secrete several hydrolytic enzymes [40]. Among the most important enzymes are aspartyl proteinases (Saps), which facilitate adhesion [36,41]. In recent years, 10 Sap genes were identified in *C. albicans* [42], 3 in *C. parapsilosis* [43], and 4 in *C. tropicalis* [44,45]. Saps only have proteinase activity in acidic environments [46]; Sap 1–3 show a specific correlation with VVC and the occurrence of *C. albicans* [47,48,49].

Phospholipases similarly play an important role in pathogenicity by damaging host cell membranes and contributing to adhesion. Mohandas and Ballal [50] observed a greater number of phospholipase-producing strains from vaginal isolates in patients with candidosis than in patients without candidosis. Moreover, different phospholipase genes (*PLA*, *PLB1–2*, *PLC1–3,* and *PLD1*) were described in infected women [51].

### 3.2. From Colonization to Infection

Vaginal colonization with *C. albicans* is common, and women with colonization are often asymptomatic [8,52]. However, when colonization progresses to infection, women frequently report vaginal itching, burning, pain, and redness. Typical VVC symptoms are often accompanied by vaginal discharge, consisting of shed epithelium, immune cells, yeast and vaginal fluid [53]. *Candida* invasion requires a transition from the yeast to the hyphae form; however, the ability to produce hyphae varies among different species [36]. In vitro studies showed that *C. albicans* without hyphae formation has a lower rate of tissue invasion [54]. In addition, the toxin candidalysin contributes to this transition, as it has a cytotoxic effect on the host cells and promotes invasion, attracting leukocytes [34,55]. Multiple virulence and host-specific factors may play a role in the development from colonization to infection.

## 4. Virulence Factors

VVC episodes cannot be attributed to a specific trigger [4,5]. The individual infection susceptibility depends on intrinsic and extrinsic factors. Host-specific risk factors, such as local defense mechanisms, age and hormonal status, pregnancy, allergies, psychosocial stress, metabolic issues, immunosuppression and individual genetic susceptibility, are important [2,56,57,58,59,60]. Additionally, behavioral risk factors, such as the use of oral contraceptives, antibiotics, glucocorticoids, inhibitors of the sodium glucose co-transporter-2 (SGLT2), intrauterine devices (IUDs), spermicides and condoms, as well as sexual, hygienic and dressing habits, need to be addressed [2,3,61].

### 4.1. Immunologic Factors

Genetic factors contribute to the development of VVC or its relapse. Foxman et al. [62] reported that women of African ethnicity showed an increased risk for VVC. This might be the result of a reduced occurrence of lactobacilli that happens more frequently in women of African ethnicity [63]. Moreover, genetic polymorphisms in blood group antigens and MBL have been identified in cases of increased susceptibility to RVVC [64,65,66,67]. A loss of the last 9 amino-acids in the carbohydrate recognition domain of the Dectin-1 gene has been associated with the occurrence of RVVC [68]. This mutation leads to insufficient production of cytokines (IL-17, tumor necrosis factor and IL-6) when it comes in contact with *Candida* (78). Moreover, women with atopic diathesis and type I allergies experience VVC more frequently than healthy individuals [69]. The typical VVC symptoms, such as itching and redness, may equally be regarded as signs of an allergic phenomenon [2,70].

Apart from common immunologic factors, pregnancy increases the likelihood of experiencing VVC; its incidence increases from 9% to 54% between the first and the third trimesters of pregnancy [71,72,73]. This rise may be partly attributed to immunologic factors, but also to the increase in sex hormones [56,57]. The occurrence of VVC during pregnancy is generally not considered dangerous with regard to preterm birth [57]. Recurrent candidosis showed an association with preterm birth in a large retrospective trial [72]. However, preterm birth is a multifactorial event, and it is likely that chronic inflammation but not VVC itself contributes to this event [72]. Of important note, almost all infants born from mothers with VVC during pregnancy show “diaper rash” or oral thrush due to the vertical mother-to-infant transmission [74].

### 4.2. Hormonal Factors

Glycogen serves as a nutrient substrate for fungi in the vaginal epithelium. There is a relationship between the respective hormonal cycle phase and the occurrence of VVC influenced by estrogen. Therefore, most women experience VVC cyclically or during the luteal phase. When estrogen levels drop during menstruation, symptoms often disappear [75]. Women who are on oral contraceptives and postmenopausal women who receive hormone replacement therapy are more likely to develop VVC than others [76]. Several studies report a higher incidence of colonization with *Candida* and VVC in women who use oral contraceptive pills [3,26,77]. The intake of contraceptives increases the level of vaginal glycogen, providing a better condition for *Candida* growth [78]. Miller et al. [79] demonstrated that IUDs compromise the vaginal defense against infections. Donders et al. [80] reported that the risk of developing VVC increased during the first 5 years after placement of levonorgestrel intrauterine systems, with the risk particularly high during the first year of placement [80]. Progesterone, however, reduces the ability of *C. albicans* to develop hyphae forms.

### 4.3. Metabolic Factors

Women with diabetes mellitus have an increased risk of experiencing VVC. This risk is exacerbated when their serum glucose levels are not within the normal range. Hyperglycemia leads to increased fungal adhesion and growth, and a glycemic index of 10–11 mmol/L can impair the host’s defense mechanisms [81]. Similarly, *C. albicans* shows a high ability to bind to vaginal epithelial cells in in vitro studies [82,83]. The yeast exhibits a glucose-inducible surface protein that promotes its adhesion to vaginal epithelial cells [81], and an increase in this protein impairs the recognition of neutrophil phagocytes [84]. Therefore, the migration of neutrophils is reduced and their functions, including phagocytosis, adhesion, chemotaxis, and intracellular killing, are impaired, increasing the sensitivity to VVC [3,5,59,80].

From a clinical perspective, women with diabetes are often unresponsive to standard antifungal treatment [58,59]. Compared with non-diabetic women, diabetic women are frequently colonized with non-*albicans* species, such as *C. glabrata* [85,86,87], impacting their treatment concept [85]. In diabetic women with RVVC, discontinuation of the antidiabetic agent should be considered [34]. The undesirable side effect of hyperglycemia caused by glucocorticoids has a similar effect to that of non-drug-induced hyperglycemia [88] because the steroid hormones suppress the immune response, increasing the susceptibility to VVC [89]. In pregnant women with gestational diabetes, their diabetic state impairs metabolic control and leukocyte function [5,80].

### 4.4. Lifestyle Factors

Various lifestyle factors can have a significant influence on the development of VVC. There is weak evidence for the impact of nutrition on *Candida* growth, although some studies have reported that the consumption of food rich in sugar and carbohydrates, as well as dairy products, can lead to increased fungal growth [90,91]. In contrast, others have reported that yogurt, oat bran and flaxseed might have positive effects in preventing fungal growth [7]. Despite their theoretical basis, these findings are not well substantiated.

Sexual behavior, especially oral sex, plays a role particularly for re-infections [92,93,94]. Reed et al. demonstrated that oral microorganisms could be transmitted from the oral cavity to the vagina [93,95,96]. Genital hygiene can also become a risk factor for VVC, when poor personal hygiene and a high frequency of sexual intercourse increase the likelihood for colonization [92], VVC [97] and RVVC [98,99]. Women should be advised to avoid excessive washing of the genital area and the use of potential irritants such as perfumed soaps, bubble baths, powders or vaginal sprays [100].

The use of tight clothing and synthetic underwear might also promote fungal growth due to the increased perineal moisture and temperature [61,97].

### 4.5. Other Exogenous Factors

Women who are colonized with *Candida* have a 33% higher risk of developing VVC after antibiotic treatment than non-colonized women [92,101,102,103]. However, the routine use of antimycotic treatment after antibiotic therapy should be avoided because it fosters drug-resistant fungi [104]. There are different theories about why VVC occurs more frequently after antibiotic treatment. One of the theories involves the reduction or eradication of vaginal and intestinal lactobacilli caused by antibiotics. As a result, affected patients lack protection from pathogenic microorganisms because lactobacilli have the ability to adhere to vaginal epithelial cells and inhibit pathogenic fungal growth [105,106].

Some lactobacilli have antagonistic effects on *Candida* [107,108]; their vaginal administration may lead to an adequate colonization and reduction in fungal load [34,109]. In vitro, lactobacilli have also shown direct fungicidal and immuno-stimulatory effects [110]. The interactions are manifold: lactobacilli can block the passage of pathogenic microbes from the gastrointestinal tract into the vagina, modulate the host’s immune response, influence epithelial defense and thus affect the expression of VVC-induced inflammatory genes. Probiotics take advantage of these effects [111].

## 5. Biofilm Formation

Microbes can form biofilms in response to various factors, including cellular recognition of specific or non-specific attachment sites on a surface, nutritional cues or exposure of planktonic cells to sub-inhibitory concentrations of antibiotics. A cell that switches to a biofilm growth mode undergoes a phenotypic shift in behavior wherein large suites of genes are differentially regulated [112]. The demonstration of a vaginal biofilm in bacterial vaginosis and its postulated importance in the pathogenesis of recurrent infections have led to the hypothesis that biofilms might be crucial for the development of VVC. However, fungal biofilms have been identified in vitro, although histological lesions in vivo are primarily polymicrobial and do not contain biofilms [106].

The shift from colonization to infection involves adherence to the vaginal epithelium, invasion, infection and inflammation through virulence factors. Most importantly, the formation of pseudo-hyphae leads to the formation of fungal components that stimulate the chemotaxis of granulocytes, causing inflammation. Changes in or overexpression of the target molecules, active extrusion of antimycotics by efflux pumps, limited diffusion of antimycotics in the matrix, stress tolerance, cell density and the presence of persistent cells may be involved in this drug-resistance phenomenon [113].

*Candida* is well known for forming biofilms on the acrylics of dentures, implantable devices in the bloodstream, urinary catheters and mucosal surfaces, such as the oral cavity [114,115]. The ability of *C. albicans*, *C. glabrata*, *C. parapsilosis*, *C. tropicalis* and *C. guilliermondii*, isolated from patients with VVC, to form biofilms, has also been investigated in vitro [116]. However, 20–34% of patients with RVVC show mixed biofilms. *Streptococcus agalactiae* and *Gardnerella vaginalis* were the bacterial microbes identified in these cases [117,118]. It remains unclear if the presence of a fungal biofilm determines whether *Candida* behaves as a pathogen or as a colonizer on the vaginal mucosa, leading to a switch from commensalism to a pathogenic state. From what is known, it can be assumed that biofilm formation causes a certain resistance to antimycotics, despite the unknown mechanisms [119].

## 6. Diagnostic Work-Up

VVC can cause psychosocial stress and may negatively affect the work and social lives of patients [60,120]. The decrease in quality of life through VVC is comparable with that of patients with bronchial asthma or chronic obstructive bronchitis [121]. In 90% of the affected women, the predominant symptom of VVC is itching. Anamnesis and proper diagnostic work-up is crucial for VVC, considering the rising number of women with similar symptoms who practice self-medication. Although many people use over-the-counter therapies, studies have shown that only 28% of self-treated women actually experienced VVC [122]. Premenopausal women experience candidosis that is mostly limited to the vestibule and vulva and symptoms that occur prior to the menstrual period, whereas postmenopausal women experience candidosis of the groin and vulvar areas. In addition, edematous labia minora and burning rhagades can occur in cases of RVVC. However, physicians should consider that only 35–40% of all women with itching complaints actually have VVC [123,124,125]. For a better differential diagnosis, it should be considered that VVC has no unpleasant odor and that the vaginal discharge of affected women is mostly of whitish color and lumpy consistency [5].

Apart from clinical presentation characteristics, the diagnosis should involve laboratory methods. Following anamnesis and gynecological examination of the patient, phase contrast microscopy should be used to examine the vaginal discharge using saline solution (or alternatively 10% KOH solution) at 400-fold magnification [126]. The clear presence of hyphae upon microscopic examination is a major criterion for appropriate diagnosis; however, it can only be observed in 50–80% of positive cases [2,126]. Microscopy is essential since the formation of hyphae plays an important role in tissue invasion [36]. Accordingly, in vitro studies have shown that *C. albicans* has a lower rate of tissue invasion in the absence of hyphae in microscopic examination [54].

In some cases, low germ load hinders microscopic detection. In particular, RVVC cases require culture tests; however, these methods should not routinely involve the determination of minimal inhibitory concentration [2,92,127,128]. The typical culture medium for detecting *Candida* spp. is Sabouraud 2% glucose agar. Chromogenic media may be superior because they are more sensitive to *Candida* and, in case of mixed cultures, certain *Candida* spp. may be immediately identified by these means. Other diagnostic methods for the detection of *Candida* are DNA hybridization tests, which have a reported sensitivity and specificity of 84.2% and 96.3% [129], or qualitative detection tests of *Candida* antigens with a sensitivity and specificity approximately 93% and 95%, respectively [130].

To date, the serological tests that determine the antibody levels of *Candida* still lack evidence and specificity. These tests may detect fungi colonization in other body parts, such as the oral cavity, in mild or superficial VVC cases.

## 7. Future Perspectives

In the future, it will be increasingly important to avoid the emergence of drug-resistant *Candida* strains, prevent cases of RVVC and consider multiple drug interactions. One of the greatest challenges will be the prevention of antimycotic drug resistances in fungal infections. Consequently, alternative treatment strategies to conventional antimycotic treatment will be increasingly important [131]. Despite the desire of many women to self-treat instead of seeking professional help [132], they should be adequately informed that over-the-counter treatment without proper diagnosis is often inaccurate and ineffective [122,133]. Moreover, decreasing the occurrence of self-treatment would assist in reducing the risk of developing drug resistance.

Because VVC constitutes a “neglected disease” in scientific research, which contributes to its relatively high prevalence, it is increasingly important to focus on its virulence factors [134]. Research on *Candida* is potentially promising as it offers various opportunities for clinical and translational studies. As an example, *Candida* vaccination could potentially become a realization [135,136,137]. Studies in animal models are ongoing, and their results are also promising [57,138].

To summarize, progress is being made in *Candida* research, but there is still more effort needed—despite the light on the horizon.

## Figures and Tables

**Figure 1 jof-06-00267-f001:**
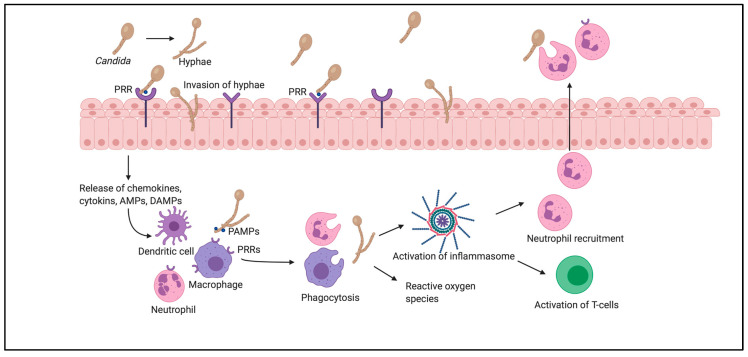
Pathogenesis of vulvovaginal candidosis.

**Table 1 jof-06-00267-t001:** Vulvovaginal candidosis (VVC) and *Candida* colonization in different countries—adapted from Gonçalves et al. [3].

			Symptomatic Women		Asymptomatic Women	
Country	Year of Study	Methods Used	Cases	VVC (%)	Cases	Colonized (%)
Australia	2003–2004	questionnaire + microscopy + culture	342	42.7	-	-
Austria	2000–2004	microscopy + culture	10,463	30.5	-	-
Brazil	2002	microscopy + culture	23	43.5	112	14.3
	2005–2007	culture	121	47.9	165	17.0
Greece	2002–2004	microscopy + culture	4743	12.1	-	-
India	2003–2004	culture	601	18.5	-	-
	-	culture	1050	20.4	-	-
	2011–2012	microscopy	300	17.7	-	-
Israel	-	microscopy + culture	208	35.5	100	15.0
Italy	1996–2005	microscopy + culture	13,014	19.5	11,551	11.6
Jamaica	-	unknown	422	29.6	-	-
Nigeria	-	culture	902	57.3	-	-
Tunisia	2006–2008	either microscopy or culture	481	48.0	-	-
Turkey	2004–2005	culture	569	42.2	-	-
**Overall**			**33,239**	**24.3**	**11,928**	**11.7**

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
