# Peer review of "Vulvovaginal Candidosis: Current Concepts, Challenges and Perspectives"

_jof, 2020, doi:10.3390/jof6040267_

Round 1

Reviewer 1 Report

I appreciate the brevity of the review that focuses on pathogenesis and diagnostics. The discussion of risk factors in light of the pathogenesis (page 4) is helpful.

Still, I think the review can be improved so that the reader gets a more complete and clearer overview. In particular, the structure of the article could be more logical in my opinion. After the section on epidemiology I would start with the pathogenesis (as described in figure 1). From colonization and yeast-to-hyphae transition is also part of this in my opinion. Subsequently, the various factors that play a role in the pathogenesis can be discussed per group. For example: immune system (genetics may be incorporated herein), aspects of the Candida involved, hormonal / metabolic factors, lifestyle and other exogenous factors. This can then be referred to when discussing the risk factors. With respect to RVVC is would be good to also include the latest view on the pathogenesis (rather autoinflammatory that immunodeficiency). See for example this recent review (DOI: 10.3390/microorganisms8020144) and original study (DOI: 10.1093/infdis/jiaa444 ).

Other remarks:

  • I suggest to include the setting in which women were tested (Table 1) to include in the table.
  • The reader would benefit from a summary of the diagnostic value (sensitivity/specificity) of the different diagnostic modalities (anamnesis, physical examination, wet mount microscopy, culture, PCR)
  • Line 36: the phrasing suggests that estrogen leads to proliferation of Candida, while the interaction is much more complex and involves increased colonization, phenotypic changes and also effect on the host leading to increased glycogen stores that is a nutrient for Candida.
  • Line 160-163: I think the certainty with which the authors state that diet has a significant influence on (R) VVC is not well enough substantiated, despite the theoretical basis for it.

Author Response

Comment: 1
I appreciate the brevity of the review that focuses on pathogenesis and diagnostics. The discussion of risk factors in light of the pathogenesis (page 4) is helpful.

Response: We would like to thank Reviewer 1 for the positive feedback on our manuscript.

Comment: 2
Still, I think the review can be improved so that the reader gets a more complete and clearer overview. In particular, the structure of the article could be more logical in my opinion. After the section on epidemiology I would start with the pathogenesis (as described in figure 1). From colonization and yeast-to-hyphae transition is also part of this in my opinion. Subsequently, the various factors that play a role in the pathogenesis can be discussed per group. For example: immune system (genetics may be incorporated herein), aspects of the Candida involved, hormonal / metabolic factors, lifestyle and other exogenous factors. This can then be referred to when discussing the risk factors.

Response: We are thankful for this important comment, which we have addressed accordingly in the revised version of our manuscript. We have adapted the structure of the article as suggested, and we believe that the reader now gets a more complete and clearer overview of the topic.

Comment: 3
With respect to RVVC is would be good to also include the latest view on the pathogenesis (rather autoinflammatory that immunodeficiency). See for example this recent review (DOI:10.3390/microorganisms8020144) and original study (DOI: 10.1093/infdis/jiaa444).

Response: We have now included these references as suggested.

Comment: 4
Other remarks: I suggest to include the setting in which women were tested (Table 1) to include in the table. The reader would benefit from a summary of the diagnostic value (sensitivity/specificity) of the different diagnostic modalities (anamnesis, physical examination, wet mount microscopy, culture, PCR).

Response: We appreciate this useful point. We have now added a separate column on the diagnostic modality that was used in Table 1.

Comment: 5
Line 36: the phrasing suggests that estrogen leads to proliferation of Candida, while the interaction is much more complex and involves increased colonization, phenotypic changes and also effect on the host leading to increased glycogen stores that is a nutrient for Candida.

Response: Indeed, we agree that the interaction of Candida growth is much more complex. This is why we have integrated a particular paragraph on the pathogenesis of the disease. To make this point comprehensible for the readership, we have changed the sentence within the paragraph on
epidemiology as follows: “Apart from asymptomatic fungal colonization, VVC is the second leading cause of vaginitis, primarily affecting women during their reproductive lifetime, when high estrogen levels increase the glycogen content of the vaginal epithelium, therewith playing a role in the nutrition for the yeast.”

Comment: 6
Line 160-163: I think the certainty with which the authors state that diet has a significant influence on (R)VVC is not well enough substantiated, despite the theoretical basis for it.

Response: This is a very important point, which we have addressed accordingly in the revised version of our manuscript, by revising the statement as follows: “There is weak evidence for the impact of nutrition on Candida growth, although some studies have reported that the consumption of food rich in sugar and carbohydrates, as well as dairy products, can lead to increased fungal
growth. In contrast, others have reported that yogurt, oat bran, and flaxseed might have positive effects in preventing fungal growth. Despite their theoretical basis, these findings are not well substantiated.”

Reviewer 2 Report

Dear editor, dear authors,

this overview is excellent and very appreciated. 

Nevertheless, I have some recommendations:

  • line 50: cite better for the frequency of vaginal Candida colonization Drell et al., PloS One 2013; 8: e54379. Relative abundance of the most abundant bacterial and fungal operational taxonomic units (OTU) found in the vaginal communities of 181 premenopausal women. They identified Candida in more than 60% in the healthy vagina!
  • - lines 61 and 160 ff, where you mention psychosocial stress: you should cite Meyer H, Goettlicher S, Mendling W. Stress as a cause of chronic recurrent vulvovaginal candidosis and the effectiveness of the conventional antimycotic therapy. mycoses 2006; 49: 202-209. This is, to the best of any knowledge, the only study to this question worldwide in cooperation of a psychologist (first author) and two gynecologists.
  • line 209 ff, biofilms: Candida biofilms have been identified in vitro. but in-vivo - so Swidsinski, Guschin, Tang et al. Vulvovaginal candidiasis: histological lesions are primarily polymicrobial and do not contain biofilms. Am J Obstet Gynecol 2018  

Author Response

Comment: 1
Dear editor, dear authors, this overview is excellent and very appreciated. Nevertheless, I have some recommendations: line 50: cite better for the frequency of vaginal Candida colonization Drell et al., PloS One 2013; 8: e54379. Relative abundance of the most abundant bacterial and fungal operational
taxonomic units (OTU) found in the vaginal communities of 181 premenopausal women. They identified Candida in more than 60% in the healthy vagina!

Response: We would like to thank you for your compliments about our manuscript. As suggested, we have added the reference of Drell et al., including the following additional sentence: “An Estonian study that used barcoded pyrosequencing technology found Candida in 67.6% of the asymptomatic women, reporting that the mycobiome that colonizes the healthy vaginal environment is more diverse than it was previously recognized.”

Comment: 2
- lines 61 and 160 ff, where you mention psychosocial stress: you should cite Meyer H, Goettlicher S, Mendling W. Stress as a cause of chronic recurrent vulvovaginal candidosis and the effectiveness of the conventional antimycotic therapy. mycoses 2006; 49: 202-209. This is, to the best of any knowledge, the only study to this question worldwide in cooperation of a psychologist (first author) and two gynecologists.

Response: Thank you. Accordingly, we have added this literature reference as suggested.

Comment: 3
line 209 ff, biofilms: Candida biofilms have been identified in vitro. but in-vivo - so Swidsinski, Guschin, Tang et al. Vulvovaginal candidiasis: histological lesions are primarily polymicrobial and do not contain biofilms. Am J Obstet Gynecol 2018

Response: We would like to thank you for this important point. Accordingly, we have added both the sentence on in-vitro biofilms of Candida and the provided reference of Swidsinski et al. to our revised manuscript, and we believe that this should now be clear for the readers of your highly esteemed journal.

Reviewer 3 Report

The present review aims to explore the main aspects of VVC. However, the different aspects are only mentioned and not exhaustively described, not giving any new point of view compared to what is already present in the literature. To be suitable for publication, the manuscript should be improved.

Minor comments:

Line 25: Please change “estrogenized vagina”

Line 25: Please, remove (spp.). It’s not necessary.

Line 26: Please, remove (C.). It’s not necessary.

Figure 1: The legend is very long. The information given in the legend should be integrated in the text.

Major comments:

Line 205-206: other lactobacilli species possess anti-Candida activity. L. rhamonosus is not the most important species inside vaginal niche. The interaction between Candida and vaginal microbiota, especially lactobacilli, should me more explained.

The potential use of probiotic in VVC treatment is only mentioned in the title (line 197) but not further explained. The conventional therapies are not clearly stated, as well as alternative and innovative strategies to deal with Candida infection, which is an emerging field.

Line 209: the part regarding biofilm is very interesting, but too hasty, and might be expanded.

Author Response

Comment: 1
The present review aims to explore the main aspects of VVC. However, the different aspects are only mentioned and not exhaustively described, not giving any new point of view compared to what is already present in the literature. To be suitable for publication, the manuscript should be improved.

Response:
Thank you for your constructive criticism. We have tried to address every comment that was made by the reviewers.

Comment: 2
Minor comments: Line 25: Please change “estrogenized vagina”

Response: We have changed this wording accordingly.

Comment: 3
Line 25: Please, remove (spp.). It’s not necessary.

Response:We have removed this abbreviation as suggested.

Comment: 4
Line 26: Please, remove (C.). It’s not necessary.

Response:We have removed this abbreviation as suggested.

Comment: 5
Figure 1: The legend is very long. The information given in the legend should be integrated in the text.

Response: We totally agree that the legend of Figure 1 was very extensive. However, as no text should be provided within the figures, we have included the text in the main body of the manuscript and shortened the figure legend accordingly.

Comment: 6

Line 205-206: other lactobacilli species possess anti-Candida activity. L. rhamnosus is not the most important species inside vaginal niche. The interaction between Candida and vaginal microbiota, especially lactobacilli, should me more explained.

Response: As suggested, we have revised the statement regarding L. rhamnosus. Moreover, we have added a separate paragraph on the interaction between lactobacilli and Candida, as follows: "Some lactobacilli have antagonistic effects on Candida; their vaginal administration may therewith lead to an adequate colonization and reduction in fungal load. In-vitro, lactobacilli have also shown direct fungicidal and immuno-stimulatory effects. The interactions are manifold: lactobacilli can block the passage of pathogenic microbes from the gastrointestinal
tract into the vagina, modulate the host's immune response, influence epithelial defense, and thus the expression of VVC-induced inflammatory genes. Probiotics take advantage of these effects."

Comment: 7
The potential use of probiotic in VVC treatment is only mentioned in the title (line 197) but not further explained. The conventional therapies are not clearly stated, as well as alternative and innovative strategies to deal with Candida infection, which is an emerging field.

Response:
Indeed, our manuscript does not contain information on the treatment of vulvovaginal candidosis. However, as stated in the Abstract of our review, we aimed to summarize the currently available knowledge on epidemiology, pathogenesis, and diagnosis of VVC. As treatment recommendations differ widely between different countries and areas of the world, we consider it important to refer to currently available guidelines regarding treatment recommendations, as we have stated in the Abstract of our manuscript. The subtitle "antibiotic
and probiotic use" does refer to the increasing risk of VVC through these measures.

Comment: 8
Line 209: the part regarding biofilm is very interesting, but too hasty, and might be expanded.

Response: We totally agree that the paragraph regarding the biofilm formation of Candida was too short. Therefore, we have significantly extended this section and hope that it is now suitable for your readers.

Round 2

Reviewer 1 Report

My comments have been well addressed

Reviewer 3 Report

The manuscript results strongly improved, and I think that it can be published in the present form.